# Ethnomedicinal Uses, Phytochemistry, and Pharmacological Activity of the *Irvingia* Species

**Branly-Natalien Nguena-Dongue** [1,†] , **Boniface Pone Kamdem** [1,2,*,†] , **Paul Keilah Lunga** [1]
**and Fabrice Fekam Boyom** [1]

1   Antimicrobial and Biocontrol Agents Unit (AmBcAU), Laboratory for Phytobiochemistry and Medicinal
    Plants Studies, Department of Biochemistry, Faculty of Science, University of Yaoundé I,
    Yaoundé P.O. Box 812, Cameroon
2   Department of Basic and Applied Fundamental Sciences, Higher Institute of Agriculture, Forestry,
    Water and Environment (HIAFWE), University of Ebolowa, Ebolowa P.O. Box 755, Cameroon
*   Correspondence: ponekamdemboniface@gmail.com; Tel.: +237-680-98-76-69 or +237-657-52-57-18
†   These authors contributed equally to this work.

**Abstract:** Plants belonging to the genus *Irvingia* are widespread across the African and Southeast Asian regions of the world. *Irvingia gabonensis*, *Irvingia malayana*, and *Irvingia grandifolia* are among the commonly used species in ethnomedicine, especially in Africa. Fever, scabies, toothache, inflammation, and liver and gastrointestinal disorders are among the pathological conditions that are reverted by *Irvingia* plants upon traditional preparations. Modern pharmacological investigations have substantiated the ethnomedicinal uses of *Irvingia* spp. Reports on the phytochemical analysis of *Irvingia* plants have revealed the presence of a number of secondary metabolites such as flavonoids, phenolic compounds, tannins, saponins, and alkaloids. Based on the foregoing, the present study provides a comprehensive evaluation of reports on the ethnomedicinal use, phytochemistry, pharmacology, and toxicity of plants from the genus *Irvingia*. Relevant information on *Irvingia* plants was mostly obtained from major scientific databases from their inception until July 2023. As a result, more than forty compounds have been identified in *Irvingia* spp., proving the abundance of secondary metabolites in these plants. Reports have pointed out modern pharmacological activities such as antiprotozoal, antimicrobial, antioxidant, antidiabetic, anti-inflammatory, and hepatoprotective activities. The present study provides more insights for the successful utilization of *Irvingia* plants and may guide further research on their therapeutic potential in the treatment of various diseases.

**Keywords:** *Irvingia* spp.; ethnomedicinal uses; phytochemistry; pharmacological activity; toxicity

## 1. Introduction

From the Irvingiaceae family, *Irvingia* (commonly called wild mango, bush mango, or ogbono) is a genus that is mainly found in African and Southeast Asian regions [1]. Taxonomically, World Flora Online has published seven accepted species of *Irvingia* that include *Irvingia malayana*, *Irvingia excelsa*, *Irvingia fusca*, *Irvingia gabonensis*, *Irvingia wombolu*, *Irvingia smithii*, and *Irvingia grandifolia* [2]. Also called bush mango or African mango, *Irvingia gabonensis* (bark) is used for the traditional treatment of dysentery, scabies, toothache, and skin diseases [3]. In combination with palm oil, the leaves of *I. gabonensis* are used to stop hemorrhage in pregnant women [4]. *Irvingia gabonensis* bark is very effective for treating skin bruises, toothaches, dysentery, and hernia [3]. The hepatoprotective activity of *Irvingia gabonensis* has also been reported by several authors [5,6]. The bark decoction of *Irvingia grandifolia* is used to relieve pain and for bathing to treat fever in children [7]. Various parts of *Irvingia grandifolia* and *Irvingia malayana* are used to treat a number of diseases such as diabetes, asthenia, icterus, dysentery, toothache, diarrhea, scabies, inflammation, and yellow fever [8–10]. Modern pharmacological properties of *Irvingia* spp. include antidiarrheal [11], antimicrobial [12–14], cytotoxic [15], antioxidant, and antidiabetic [16,17] activities, among

others. The phytochemical screening of *Irvingia* spp. revealed the presence of saponins, tannins, alkaloids, flavonoids, cardiac glycosides, steroids, carbohydrates, volatile oils, and terpenoids [3] as well as phenolic acids such as 2,3,8-tri-O-methylellagic acid [18]. Regarding the phytochemistry and pharmacological studies on *Irvingia* spp., a number of studies are available across the literature. However, few reviews have been reported on *Irvingia* plants. In fact, emphasis has been placed mostly on food applications [19] and the cardiovascular disease outcomes [20,21] of *Irvingia* species. Thus, comprehensive reviews and/or monographs on the application of *Irvingia* species as ethnomedicinal plants are needed. In this line, the present work aims to summarize existing information regarding the phytochemistry, pharmacological activities, and ethnomedicinal uses of *Irvingia* plants.

## 2. Research Methods

The present study aims to substantially review the phytochemistry, pharmacological activity, and traditional uses of *Irvinvia* plants.

### 2.1. Literature Search

Information related to the pharmacological activity of chemical constituents from *Irvingia* plants was obtained from published and unpublished materials across the literature. Relevant data were obtained from databases such as PubMed (National Library of Medicine), American Chemical Society (ACS), Science Direct, SciFinder, Web of Science, Scopus, Wiley, Google Scholar, and Springer from their respective inception until August 2023. Furthermore, theses, dissertations, and textbooks were searched to obtain the relevant data. The search terms included "*Irvingia*"; OR "*Irvingia* spp." AND "Traditional uses"; "*Irvingia*"; OR "*Irvingia* spp." AND "Phytochemistry"; "*Irvingia*" OR "*Irvingia* spp." AND "Pharmacology"; "*Irvingia*" OR "*Irvingia* spp." AND "Pharmacological activity"; "*Irvinbgia*" OR "*Irvingia* spp." AND "Toxicity". In addition, books, scientific reports, dissertations, theses, and articles published in peer-reviewed journals were also scrutinized and searched. References obtained from various searches were also examined to obtain further additional information.

### 2.2. Data Extraction and Synthesis

Data extraction and synthesis were conducted by the first and second authors and confirmed by the other authors. Figures or tables were used to gather appropriate data that were further summarized and analyzed. In addition, a descriptive narration was used to provide a summary of the results. The chemical structures of the pharmacologically active compounds are provided using graphical representations.

### 2.3. Results of the Literature Search

A total of 1842 ("*Irvingia*": 805; "*Irvingia* and traditional uses": 345; *Irvingia* and Phytochemistry": 45; *Irvingia* and Pharmacological activity": 193; "*Irvingia* and Pharmacology": 182; "*Irvingia* and Toxicity": 272) significant records were obtained, of which 630 were excluded for data duplicates. Among the 1212 records obtained after removing the duplicates, 1165 (Figure 1) were disqualified and omitted after screening the titles or abstracts. Thus, forty seven (47) full-length research articles were identified, of which seven were excepted as they were review articles. Finally, a total of forty (40) full text articles were included and exploited to collect relevant information. In addition, important facts obtained from unpublished materials were also included.

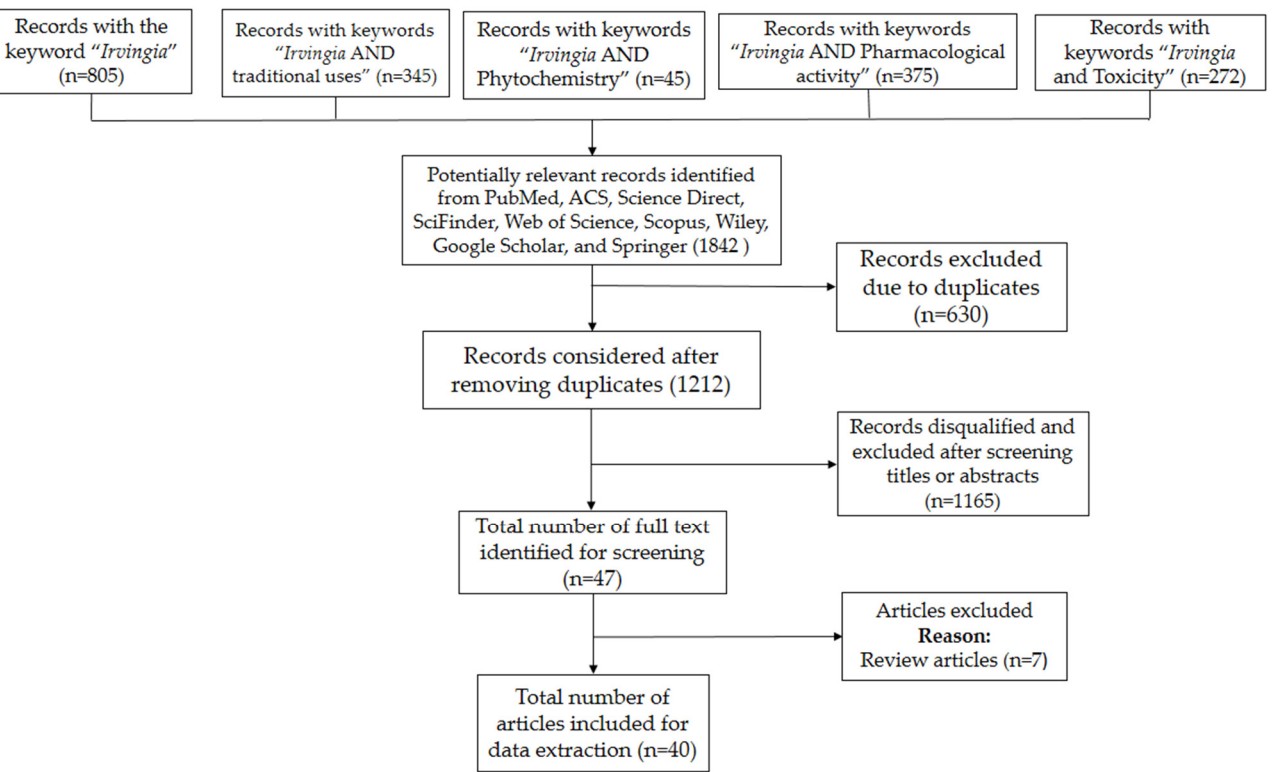

**Figure 1.** Flowchart of the literature search and selection procedure.

### 3. Phytochemistry of *Irvingia* spp.

Previous phytochemical screening of *Irvingia gabonensis* and *Irvingia wombolu* peels, seed coats, leaves, and seeds revealed the presence of flavonoids, alkaloids, phenolic compounds, tannins, steroids, and saponins [13,22]. More importantly, previous reports pointed out the isolation and identification of the following classes of compounds: terpenoids, flavonoids, phenolic compounds, and miscellaneous compounds.

*3.1. Terpenoids*

A couple of research groups [23,24] isolated and characterized several terpenoids from *Irvingia gabonensis*. These include Kuete et al. [23], who separated four terpenoids [friedelanone (138.88 μg/g of extract) (**1**), betulinic acid (3.31 mg/g of extract) (**2**), oleanolic acid (5.09 mg/g of extract) (**3**), and hardwickiic acid (11.95 mg/g of extract) (**4**)] from the stem bark of *Irvingia gabonensis*. In addition to the diterpenoid hardwickiic acid (**4**), Donfack et al. [24] isolated a triterpenoid, namely, 3-β-acetoxyursolic acid (**5**) (Table 1). From *Irvingia malayana*, Ng et al. [25] isolated and characterized betulinic acid (**2**), a compound that was already identified in *Irvingia gabonensis* by Kuete et al. [23] and Donfack et al. [24].

*3.2. Flavonoids*

Regarding advances on the isolation of flavonoids from *Irvingia* species, a few flavonoids have been previously separated and characterized. These includes five flavonols [kaempferol 3-O-glucoside (**6**), quercetin 3-O-rhamnoside (**7**), rhamnetin (**8**) (from seeds) [26], quercetrin (**9**), kaempferol (**10**) (from stem bark) [27]] and two flavones [diosmetin (**11**) [28] and apigenin (**12**)] [27] from the seeds and stem bark, respectively (Table 2).

**Table 1.** Terpenoids isolated thus far from *Irvingia* spp.

| Serial Number | Phytochemical Classification | Compound Name | Chemical Structure | Plant Species | Plant Organs | References |
|---|---|---|---|---|---|---|
| **1** | Triterpenoid | 3-Friedelanone |  | *Irvingia gabonensis* | Stem bark | [23,24] |
| **2** | Triterpenoid | Betulinic acid |  | *Irvingia gabonensis; Irvingia malayana* | Stem bark | [23–25] |
| **3** | Triterpenoid | Oleanolic acid |  | *Irvingia gabonensis* | Stem bark | [23,24] |

**Table 1.** *Cont.*

| Serial Number | Phytochemical Classification | Compound Name | Chemical Structure | Plant Species | Plant Organs | References |
|---|---|---|---|---|---|---|
| **4** | Diterpenoid | Hardwickiic acid |  | *Irvingia gabonensis* | Stem bark | [23,24] |
| **5** | Pentacyclic triterpenoid | 3-β-Acetoxyursolic acid |  | *Irvingia gabonensis* | Stem bark | [24] |

**Table 2.** Flavonoids isolated thus far from *Irvingia* spp.

| Serial Number | Phytochemical Classification | Compound Name | Chemical Structure | Plant Species | Plant Organs | Reference |
|---|---|---|---|---|---|---|
| 6 | Flavonol | Kaempferol 3-O-glucoside |  | *Irvingia gabonensis* | Seeds | [26] |
| 7 | Flavonol | Quercetin 3-O-rhamnoside |  | *Irvingia gabonensis* | Seeds | [26] |
| 8 | Flavone | Rhamnetin |  | *Irvingia gabonensis* | Seeds | [26] |

**Table 2.** *Cont.*

| Serial Number | Phytochemical Classification | Compound Name | Chemical Structure | Plant Species | Plant Organs | Reference |
|---|---|---|---|---|---|---|
| 9 | Flavone | Diosmetin | | *Irvingia gabonensis* | Seeds | [26] |
| 10 | Flavonol | Quercetrin | | *Irvingia gabonensis* | Stem bark | [27] |
| 11 | Flavonol | Kaempferol | | *Irvingia gabonensis* | Stem bark | [27] |
| 12 | Flavone | Apigenin | | *Irvingia gabonensis* | Stem bark | [27] |

### 3.3. Phenolic Compounds

Various separation techniques (column chromatography, ultra-high-performance liquid chromatography, etc.) have been used to isolate or identify phenolic compounds from plants of the genus *Irvingia*. In 2012, Sun and Chen [26] used UHPLC/HRMS (ultra-high-performance liquid chromatography/high resolution mass spectrometry) analysis to identify the chemical constituents of *Irvingia gabonensis* seeds. Among the compounds identified, phenolics comprised: 3,3′,4′-tri-O-methylellagic acid (4.04 mg/g of extract) (**13**); 3,4-di-O-methylellagic acid (0.58 mg/g of extract) (**14**); methyl gallate (**15**); ellagic acid (**16**), di-O-methyl-ellagic acid hexoside (**17**); methyl-ellagic acid (**18**); mono-O-methyl ellagic acid deoxyhexoside (**19**); di-O-methyl ellagic acid (**20**); di-O-methyl ellagic acid-O-pentoside (**21**); di-hexahydroxydiphenoyl-ellagic acid (**22**); di-O-methyl-ellagic acid deoxyhexoside (**23**); galloyl-hexahydroxydiphenoyl-methyl-ellagic acid (**24**); tri-O-methyl-ellagic acid (**25**); mono-O-methyl-ellagic acid rhamnoside (**26**); mono-O-methyl-ellagic acid rhamnosyl-rhamnoside (**27**); galloyl-tri-O-methyl-ellagic acid hexoside (**28**); galloyl-HHDP-ellagic acid (**29**); di-O-methyl-ellagic acid deoxyhexide (**30**) [26] (Table 3). From the powdered seeds of African mango (*I. gabonensis*), 4-hydroxybenzoic acid (**31**) was also isolated and characterized. Furthermore, ellagic acid (**16**) was also identified in *Irvingia gabonensis* stem bark by Ojo et al. [27]. More recently, Yoon et al. [29] isolated the phenolic compound (tannin) terminalin (**32**) from *Irvingia gabonensis* seeds.

### 3.4. Miscellaneous Compounds

A total of thirteen miscellaneous compounds were identified from *Irvingia* species in former studies [26,31]. These include a series of organic acids [citric acid (1R, 2S) (**33**), citric acid (1R, 2R) (**34**), citric acid (1S, 2S) (**35**), and citric acid (1S, 2R) (**36**)] that were isolated from *Irvingia gabonensis* seeds [28]. In addition to these compounds, two sugars [hexahydroxydiphenoyl(HHDP)-hexose (**37**) and di-hexahydroxydiphenoyl-hexose (**38**)] were identified [28]. Moreover, three carboxylic acids [methyl 2-[2-formyl-5-(hydroxymethyl)-1H-pyrrol-1-yl]-propanoate (**39**), 4-[formyl-5-(methoxymethyl)-1H-pyrrol-1-yl]butanoic acid (**40**), and methyl-5-hydroxy-2-pyridinecarboxylate (**41**)], three aldehydes [5-(methoxymethyl)-1H-pyrrole-2-carbaldehyde (**42**), 5-(hydroxymethyl)-1H-pyrrole-2-carbaldehyde (**43**), and 5-hydroxymethyl-2-furancarbaldehyde (**44**)], and one ketone [5-hydroxy-2-pyridyl methyl ketone (**45**)] (Table 4) were identified from *Irvingia gabonensis* seeds by Li et al. [31].

**Table 3.** Phenolic compounds isolated or identified thus far from *Irvingia* plants.

| Serial Number | Phytochemical Classification | Compound Name | Chemical Structure | Plant Species | Plant Organs | References |
|---|---|---|---|---|---|---|
| **13** | Phenolic | 3,3′,4′-tri-O-methylellagic acid | | *Irvingia gabonensis* | Stem bark, seeds, | [23,24,26] |
| **14** | Phenolic | 3,4-di-O-methylellagic acid | | *Irvingia gabonensis* | Stem bark | [23,24] |
| **15** | Phenolic acid | Methyl gallate | | *Irvingia gabonensis* | Seeds, | [24,30] |
| **16** | Phenolic | Ellagic acid | | *Irvingia gabonensis* | Seeds, stem bark | [26,27] |

**Table 3.** *Cont.*

| Serial Number | Phytochemical Classification | Compound Name | Chemical Structure | Plant Species | Plant Organs | References |
|---|---|---|---|---|---|---|
| **17** | Ellagitannin | di-O-Methyl-ellagic acid hexoside | | *Irvingia gabonensis* | Seeds | [26] |
| **18** | Phenolic | Methyl-ellagic acid | | *Irvingia gabonensis* | Seeds | [26] |
| **19** | Phenolic | Mono-O-methyl ellagic acid deoxyhexoside | | *Irvingia gabonensis* | Seeds | [26] |

**Table 3.** *Cont.*

| Serial Number | Phytochemical Classification | Compound Name | Chemical Structure | Plant Species | Plant Organs | References |
|---|---|---|---|---|---|---|
| **20** | Phenolic | Di-O-methyl ellagic acid | | *Irvingia gabonensis* | Seeds | [26] |
| **21** | Phenolic | Di-O-methyl ellagic acid-O-pentoside | | *Irvingia gabonensis* | Seeds | [26] |
| **22** | Phenolic | Di-hexahydroxydiphenoyl-ellagic acid | | *Irvingia gabonensis* | Seeds | [26] |

**Table 3.** *Cont.*

| Serial Number | Phytochemical Classification | Compound Name | Chemical Structure | Plant Species | Plant Organs | References |
|---|---|---|---|---|---|---|
| **23** | Phenolic | di-O-Methyl-ellagic acid deoxyhexoside |  | *Irvingia gabonensis* | Seeds | [26] |
| **24** | Phenolic | Galloyl-hexahydroxydiphenoyl-methyl-ellagic acid |  | *Irvingia gabonensis* | Seeds | [26] |
| **25** | Phenolic | Tri-O-methyl-ellagic acid |  | *Irvingia gabonensis* | Seeds | [26] |

<div align="center"><b>Table 3.</b> <i>Cont.</i></div>

| Serial Number | Phytochemical Classification | Compound Name | Chemical Structure | Plant Species | Plant Organs | References |
|---|---|---|---|---|---|---|
| **26** | Phenolic | Mono-O-methyl-ellagic acid rhamnoside |  | *Irvingia gabonensis* | Seeds | [26] |
| **27** | Phenolic | Mono-O-methyl-ellagic acid rhamnosyl-rhamnoside |  | *Irvingia gabonensis* | Seeds | [26] |
| **28** | Phenolic | Galloyl-tri-O-methyl-ellagic acid hexoside |  | *Irvingia gabonensis* | Seeds | [26] |

**Table 3.** *Cont.*

| Serial Number | Phytochemical Classification | Compound Name | Chemical Structure | Plant Species | Plant Organs | References |
|---|---|---|---|---|---|---|
| **29** | Ellagitannin | Galloyl-HHDP-ellagic acid |  | *Irvingia gabonensis* | Seeds | [26] |
| **30** | Ellagitannin | Di-O-methyl-ellagic acid deoxyhexoside |  | *Irvingia gabonensis* | Seeds | [26] |

**Table 3.** *Cont.*

| Serial Number | Phytochemical Classification | Compound Name | Chemical Structure | Plant Species | Plant Organs | References |
|---|---|---|---|---|---|---|
| **31** | Phenolic | 4-Hydroxybenzoic acid | | *Irvingia gabonensis* | Seeds | [26] |
| **32** | Tannin | Terminalin | | *Irvingia gabonensis* | Seeds | [29] |

**Table 4.** Miscellaneous compounds identified thus far from plants of the genus *Irvingia*.

| Serial Number | Phytochemical Classification | Compound Name | Chemical Structure | Plant Species | Plant Organs | Reference |
|---|---|---|---|---|---|---|
| **33** | Organic acid | Citric acid (1R, 2S) | | *Irvingia gabonensis* | Seeds | [26] |
| **34** | Organic acid | Citric acid (1R, 2R) | | *Irvingia gabonensis* | Seeds | [26] |
| **35** | Organic acid | Citric acid (1S, 2S) | | *Irvingia gabonensis* | Seeds | [26] |
| **36** | Organic acid | Citric acid (1S, 2R) | | *Irvingia gabonensis* | Seeds | [26] |
| **37** | Sugar (ose) | Hexahydroxydiphenoyl(HHDP)-hexose | | *Irvingia gabonensis* | Seeds | [26] |

**Table 4.** *Cont.*

| Serial Number | Phytochemical Classification | Compound Name | Chemical Structure | Plant Species | Plant Organs | Reference |
|---|---|---|---|---|---|---|
| **38** | Sugar (ose) | di-Hexahydroxydiphenoyl-hexose |  | *Irvingia gabonensis* | Seeds | [26] |
| **39** | Carboxylate | Methyl 2-[2-formyl-5-(hydroxymethyl)-1H-pyrrol-1-yl]-propanoate |  | *Irvingia gabonensis* | Seeds | [31] |
| **40** | Carboxylic acid | 4-[Formyl-5-(methoxymethyl)-1H-pyrrol-1-yl]butanoic acid |  | *Irvingia gabonensis* | Seeds | [31] |
| **41** | Carboxylate | Methyl-5-hydroxy-2-pyridinecarboxylate |  | *Irvingia gabonensis* | Seeds | [31] |

**Table 4.** *Cont.*

| Serial Number | Phytochemical Classification | Compound Name | Chemical Structure | Plant Species | Plant Organs | Reference |
|---|---|---|---|---|---|---|
| **42** | Carbaldehyde | 5-(Methoxymethyl)-1H-pyrrole-2-carbaldehyde | | *Irvingia gabonensis* | Seeds | [31] |
| **43** | Carbaldehyde | 5-(Hydroxymethyl)-1H-pyrrole-2-carbaldehyde | | *Irvingia gabonensis* | Seeds | [31] |
| **44** | Carbaldehyde | 5-Hydroxymethyl-2-furancarbaldehyde | | *Irvingia gabonensis* | Seeds | [31] |
| **45** | Ketone | 5-Hydroxy-2-pyridyl methyl ketone | | *Irvingia gabonensis* | Seeds | [31] |

## 4. Pharmacological Activity of *Irvingia* Plants

### 4.1. Antiprotozoal Activity

As protozoan diseases are a major threat to human health, medicinal plants have long been used to cure such ailments. It has also been reported that such plants possess a number of antiprotozoal hit compounds. Various organs of *Irvingia gabonensis* (antiplasmodial and anti-*Trypanosoma brucei*), *Irvingia malayana* (antiplasmodial), and *Irvingia grandifolia* (antileishmanial) have been reported to exhibit antiprotozoal activity (Table 5). In fact, studies by Atindehou et al. [28] have demonstrated the effectiveness of the crude ethanol extract of *Irvingia gabonensis* stem bark against *Plasmodium falciparum* strain K1 (resistant to chloroquine and pyrimethamine) and *Trypanosoma brucei* rhodesiense with $IC_{50}$ values of 8 and >5 μg/mL vs. suramin ($IC_{50}$: 0.010 μg/mL) and chloroquine ($IC_{50}$: 0.064 μg/mL), respectively [28]. In 2007, Nguyen-Pouplin et al. [32] obtained $IC_{50}$ values of 5.0 μg/mL and 10.5 μg/mL for methanol and ethanol extracts of *Irvingia malayana*, respectively (vs. chloroquine; $IC_{50}$: 0.1 μM), upon testing against the chloroquine-resistant FcB1/Colombia strain of *Plasmodium falciparum*. Regarding cytotoxicity experiments, methanol and ethanol extracts were not cytotoxic toward HeLa cells ($IC_{50}$: 14.8 and 11.7 μg/mL; SI: 2.9 and 1.1, respectively). Against MRC5 cells, the methanol extract yielded an $IC_{50}$ value of 50.5 μg/mL (SI: 10) [32].

In an *in vivo* study, Agubata et al. [33] revealed the antiplasmodial activity of *Irvingia gabonensis* fats and homolipid-based artemether microparticles combined with *Irvingia gabonensis* in *Plasmodium berghei*-infected mice. In fact, the oral administration of *Irvingia gabonensis* fat (AD) and microparticle capsules of artemether (4 mg/kg) combined with I. gabonensis lipid matrices (LM) and phospholipon1 90 G (P90G) [ADP3 (3:1), ADP4 (4:1), and ADP9 (9:1)] led to a significant inhibition of the parasite (percent inhibition: 83.84, 83.68, 82.63, and 87.37%, respectively) compared to the artemether treatment (percent inhibition: 56.32%) [33]. Furthermore, Lamidi et al. [34] demonstrated the *in vitro* antileishmanial activity of methanol, methanol/water (50:50), and dichloromethane extracts from the stem bark and leaves with a common $IC_{50}$ value (100 μg/mL). However, the methanol, methanol/water (50:50), and dichloromethane extracts from the stem bark ($IC_{50}$ values: 7.7, 8.4, and 8.3 μg/mL, respectively) and leaves ($IC_{50}$ values: >100, 6.2 and 5.4 μg/mL, respectively) of *I. grandifolia* were cytotoxic when tested against Vero cells [34]. Previous studies have demonstrated the presence of steroids, flavonoids, alkaloids, cardiac glycosides, volatile oils, terpenoids, tannins, saponins, etc. in *Irvingia* plants [35].

Secondary metabolites such as flavonoids have been proven to inhibit the growth of several protozoan parasites including *Trypanosoma* spp. and *Leishmania* spp. [36]. In fact, flavonoids are capable of binding to the C-terminal nucleotide-binding domain of the P-glycoprotein-like transporter in *Leishmania* spp. (a transporter involved in parasite multidrug resistance). This class of secondary metabolites works as inhibitors of enzymes or proteins that are crucial for the survival and virulence of certain Trypanosomatidae. These enzymes include DNA topoisomerases, protein tyrosine kinase, and squalene synthase. Flavonoids have been shown to induce the apoptosis of host cells such as epithelial cells [37] or by the direct induction of the apoptosis of the parasite [38,39].

**Table 5.** Antiprotozoal activity of plants from the genus *Irvingia*.

| *Irvingia* spp. | Extracts/ Compounds | Model of the Study | Significant Results | Toxicity/ Cytotoxicity | Type of Screening | Reference |
|---|---|---|---|---|---|---|
| *Irvingia gabonensis* | Crude ethanol extract of the stem bark. | -*Plasmodium falciparum* strain K1; -*Trypanosoma brucei rhodesiense*. | -$IC_{50}$: >5 µg/mL, vs. chloroquine ($IC_{50}$: 0.064 µg/mL); -$IC_{50}$: 8 µg/mL, vs. suramin ($IC_{50}$: 0.010 µg/mL). | NS | Alamar Blue assay | [28] |
| *Irvingia malayana* Oliv. ex Benn. | Methanol and ethanol (80%) extract from the leaves. | -Chloroquine-resistant FcB1/Colombia strain of *Plasmodium falciparum*; -Cytotoxicity: The human cervix carcinoma cells (HeLa), and the human diploid embryonic lung cells (MRC5). | $IC_{50}$: 5.0 and 10.5 µg/mL for methanol and ethanol extracts, respectively, vs. chloroquine ($IC_{50}$: 0.1 µM) | Cytotoxicity<br><br>• Methanol extract<br>-HeLa cells; $IC_{50}$: 14.8 µg/mL and SI: 2.9;<br>-MRC5 cells: $IC_{50}$: 50.5 µg/mL and SI: 10.0.<br><br>• Ethanol extract<br>-HeLa cells; $IC_{50}$: 11.7 µg/mL and SI: NT;<br>-MRC5 cells: $IC_{50}$: 1.1 µg/mL and SI: NT. | Antiplasmodial test: [3H]Hypoxanthine Uptake Assay; -Cytotoxicity: methyl thiazole tetrazolium (MTT) assay | [32] |
| *Irvingia gabonensis* var. excelsa (*Irvingia wombolu*) | Microparticles prepared from fatty acids from the nuts of *Irvingia gabonensis*, extracted using petroleum ether (microparticles composed of artemether (4 mg/kg), lipid matrices (LM) and phospholipon1 90 G (P90G) [ADP3 (3:1), ADP4 (4:1) and ADP9 (9:1)]). | *Plasmodium berghei*-infected mice (*in vivo*). | Percentage of inhibition of *Plasmodium berghei*: 83.84, 83.68, 82.63, and 87.37% for ADP3 (3:1), ADP4 (4:1), and ADP9 (9:1), respectively, vs. artemether treatment alone (percent inhibition: 56.32%) inhibition: 56.32%). | NS | Four day suppressive test | [33] |

**Table 5.** *Cont.*

| *Irvingia* spp. | Extracts/ Compounds | Model of the Study | Significant Results | Toxicity/ Cytotoxicity | Type of Screening | Reference |
|---|---|---|---|---|---|---|
| *Irvingia grandifolia* | Methanol, methanol/water (50:50), and dichloromethane extracts prepared from stem barks and leaves. | Promastigote form of *Leishmania infantum*; Cytotoxicity on human cell line (Vero cells) (*in vitro*). | $IC_{50}$: >100 µg/mL for the extracts assayed. | Cytotoxicity -Stem bark: $IC_{50}$: 7.7, 8.4 and 8.3 µg/mL respectively for the extracts, respectively; -Leaves: $IC_{50}$: >100, 6.2, and 5.4 µg/mL for the extracts, respectively. | Cell viability test | [34] |

ADP: Arthemether-Dika (*Irvingia*) fat-Phospholipon; HeLa: human cervix carcinoma cells; $IC_{50}$: half maximal inhibitory concentration; LM: lipid matrix; MRC5: human diploid embryonic lung cells; MTT: 3-(4,5-dimethylthiazol-2-yl)-2,5-diphenyltetrazolium bromide; NS: not specified; NT: not tested; P90G: phospholipon1 90 G; SI: selectivity index.

## 4.2. Antimicrobial Activity

According to the literature, numerous studies have demonstrated the antimicrobial activity of plants from the genus *Irvingia* (Table 6). One such study is that of Osadebe and Ukwueze [40], who reported the antibacterial activity (MIC values: 4.31, 3.53, 5.53, and 5.55 µg/mL, respectively) of a petroleum ether extract from *Irvingia gabonensis* leaves against *Staphylococcus aureus*, *Bacillus subtilis*, *Pseudomonas aeruginosa*, and *Salmonella typhi*. Moreover, Kuete et al. [23] evaluated the *in vitro* antimicrobial activity of the methanol extract, fractions (A, B, and C), and compounds [3-friedelanone (**1**), betulinic acid (**2**), oleanolic acid (**3**), hardwickiic acid (**4**), 3,3′,4′-tri-O-methylellagic acid (**13**), and 3,4-di-O-methylellagic acid (**14**)] from the stem bark of *Irvingia gabonensis* on 22 species of microorganisms including bacteria [*Escherichia coli*, *Klebsiella pneumoniae*, *Morganella morganii*, *Neisseria gonorrhoeae*, *Salmonella typhi*, *Streptococcus faecalis*, *Staphylococcus aureus*, *Citrobacter freundii*, two *Enterobacter* species, three *Proteus* species, two *Shigella* species, and four *Bacillus* species (MIC range: 1.22–312.50 µg/mL), vs. gentamicin (MIC range: 0.61–9.76 µg/mL)] and fungi [*Candida albicans*, *Candida krusei*, and *Candida glabrata* (MIC range: 9.76–312.50 µg/mL), vs. nystatin (MIC range: 2.44–9.76 µg/mL)] [23]. Furthermore, *Candida albicans* was also inhibited (MIC value: 11.4 µg/mL) by the ethanol extract of *Irvingia malayana*. Interestingly, this extract showed an MIC value greater than 100 µg/mL when tested against Vero cells, attesting to its high selectivity (SI: >8.77) [41]. Two years later, Nworie et al. [42] evaluated the *in vitro* antibacterial activity of hot water, cold water, and ethanol extracts of *Irvingia gabonensis* leaves and stem bark on *Staphylococcus aureus* and *Escherichia coli* using agar-well diffusion and agar dilution methods. As a result, the diameter of inhibition ranged between 8 and 23 mm for the ethanol extract, 8–14 mm for the hot water extract, and 8–20 mm for the cold water extract. Moreover, the MIC values ranged between 6.25 and 50 mg/mL [42]. Aqueous, methanol, and hexane extracts extracted from *Irvingia gabonensis* leaves were administered to *Escherichia coli*-induced diarrheal mice at 100 and 200 mg/kg. As a result, the treatment significantly reduced the mouse pathogenic conditions by 80 and 60%, 80 and 80%, and 60 and 40% for the aqueous, methanol, and hexane extracts, respectively [43] (Table 6). Wamba et al. [44] demonstrated the anti-staphylococcal activity of the methanol extract of *Irvingia gabonensis* leaves against seven (**7**) clinical isolates, namely SA18, SA23, SA56, SA116, MRSA3, MRSA9, and MRSA11, with an MIC value of 1024 µg/mL [44]. In addition, Olanrewaju et al. [13] assessed the antibacterial activity of the chloroform extract of *I. gabonensis* leaves against a series of bacteria (*Salmonella typhi*, *Klebsiella pneumoniae*, *Escherichia coli*, and *Pseudomonas aeruginosa*) and fungi (*Trichophyton rubrum* and *Candida albicans*). Interestingly, the chloroform extract inhibited the growth of *S. aureus* and *S. typhi* (MIC: 0.625 mg/mL), *K. pneumoniae* and *E. coli* (MIC: 5 mg/mL), *P. aeruginosa* and *S. paratyphi* (MIC: 10 mg/mL), *T. rubrum* (MIC: 10 mg/mL) and *C. albicans* (MIC: 2.5 mg/mL) [13].

More recently, Wandji et al. [45] reported the inhibitory effect of the methanol extract of *I. gabonensis* seeds against *Staphylococcus aureus* and *Fusarium oxysporum* with MIC values of 2 and 6.5 mg/mL, respectively [45].

As reported by Kuete et al. [23], *Irvingia* spp. possess a variety of antimicrobial compounds such as the triterpenoid 3-friedelanone (**1**) and the phenolic acids betulinic acid (**2**), oleanolic acid (**3**), 3,3′,4′-tri-O-methylellagic acid (**13**), 3,4-di-O-methylellagic acid (**14**), and hardwickiic acid (**4**) (Table 6). Generally, triterpenoids are reported to exhibit a number of biological activities such as antibacterial and antifungal activities. Several reports have revealed the inhibitory potential of triterpenoids on the expression of bacterial genes that are associated with the formation of biofilms. Cell autolysis and peptidoglycan cell wall turnover are also affected by triterpenoids [46]. Other mechanisms of action of this class of compounds include DNA fragmentation, cell cycle arrest [47], and apoptosis [48]. Antimicrobial mechanistic studies revealed that phenolic compounds might induce cell membrane destruction or suppress the formation of biofilms. Phenolics are also known to inhibit the virulence factors in bacteria [49].

**Table 6.** Antimicrobial activity of the extracts and compounds from plants of the genus *Irvingia*.

| *Irvingia* spp. | Extracts/Compounds | Model of the Study | Significant Results | Toxicity/Cytotoxicity | Type of Screening | Reference |
|---|---|---|---|---|---|---|
| *Irvingia gabonensis* | Methanol extract, fractions (A, B, and C), and compounds [3-friedelanone (**1**), betulinic acid (**2**), oleanolic acid (**3**), 3,3′,4′-tri-O-methylellagic acid (**13**), 3,4-di-O-methylellagic acid (**14**) and hardwickiic acid (**4**)] from the stem bark. | 22 species of microorganisms (*in vitro*). | MIC values:<br>-Fungi:<br>*Candida albicans* (MIC values: 39.06–312.50 μg/mL); *Candida glabrata* (MIC values: 19.53–312.50 μg/mL); *Candida krusei* (MIC values: 9.76–312.50 μg/mL), vs. nystatin (2.44–9.76 μg/mL);<br>-Bacteria:<br>*Klebsiella pneumoniae*, *Morganella morganii*, *Neisseria gonorrhoeae*, *Salmonella typhi*, *Streptococcus faecalis*, *Staphylococcus aureus*, *Citrobacter freundii*, two *Enterobacter* species, three *Proteus* species, two Shigella species, and four *Bacillus* species; MIC range: 1.22–312.50 μg/mL), vs. gentamicin (MIC range: 0.61 μg/mL–9.76 μg/mL) | NS | Minimum inhibitory concentration (MIC) assays. | [23] |
| *Irvingia gabonensis* | Leaves/petroleum ether extract. | *Staphylococcus aureus*, *Bacillus subtilis*, *Pseudomonas aeruginosa* and *Salmonella typhi* (*in vitro*). -Strains | MIC values: 4.31, 3.53, 5.53, and 5.55 μg/mL for the strains, respectively. | NS | Minimum inhibitory concentration (MIC) assays. | [40] |
| *Irvingia malayana Oliver ex Bennett* | Ethanol extract | *Mycobacterium tuberculosis* H37Rv, *Staphylococcus aureus*, *Escherichia coli*, and *Candida albicans*. -Cytotoxicity: Vero cells (*in vitro*). | Percentage of inhibition at the concentration of 11.4 μg/mL: 19%, 31%, 38%, and 98% against *Mycobacterium tuberculosis* H37Rv, *Staphylococcus aureus*, *Escherichia coli*, and *Candida albicans*, respectively. | Ethanol extract: IC$_{50}$: 100 μg/mL on Vero cells; SI: >8.77. | Alamar Blue assay. | [41] |
| *Irvingia gabonensis* | Hot water, cold water and ethanol extracts prepared from the leaves and stem bark. | *Staphylococcus aureus* and *Escherichia coli* (*in vitro*). | -Diameter of inhibition: ranging between 8 and 23 mm, 8–14 mm, and 8–20 mm for the extract, respectively;<br>-MIC range: 6.25–50 mg/mL. | NS | Agar-well diffusion and agar dilution methods. | [42] |
| *I. gabonensis* | Methanol extract of the seeds | *Staphylococcus aureus* and *Fusarium oxysporum* (*in vitro*). | MIC values: 2 and 6.5 mg/mL against the tested strains, respectively. | NS | Minimum inhibitory concentration (MIC) assays. | [45] |

IC$_{50}$: half maximal inhibitory concentration; *I. gabonensis*: *Irvingia gabonensis*; MIC: minimum inhibitory concentration; NS: not specified; SI: selectivity index.

*4.3. Antioxidant Activity/Capacity*

The majority of diseases are intricately related to redox imbalance and oxidative stress. The inhibition of this phenomenon by plant secondary metabolites has been proven to be efficient in reducing the pathogenesis of several diseases. As reported by many authors, the presence of a variety of compounds such as flavonoids, phenolic compounds, alkaloids, tannins, and saponins, among others, in medicinal plants has contributed to reverting oxidative stress in disease pathogenesis. One such plant includes *Irvingia gabonensis*, which has been reported to exhibit antioxidant activity or to provide protection against damage caused by free radicals (antioxidant capacity). In brief, Agbor et al. [50] used Folin–Ciocalteu reagent (Folin) and ferric reducing antioxidant power (FRAP) tests to evaluate the total antioxidant capacity of a methanol extract from *Irvingia gabonensis* seeds [50]. In the FRAP assay, the free and total antioxidant capacities were found to be 283.1 and 431.58 mg/g, respectively. In the Folin test, the free and total antioxidant capacities were found to be 7.26 and 10.74 mg/g, respectively. In another experiment, Arogba and Omede [51] described the antiradical scavenging activity of the methanol extract of *Irvingia gabonensis* seeds with an $IC_{50}$ value of 177.22 µg/mL compared with the activity of vitamin C ($IC_{50}$: 300.22 µg/mL) and quercetin (184.71 µg/mL) [51]. Three compounds [methyl 2-[2-formyl-5-(hydroxymethyl)-1H-pyrrol-1-yl]-propanoate (**39**), 4-[formyl-5-(methoxymethyl)-1H-pyrrol-1-yl]butanoic acid (**40**), and 5-hydroxy-2-pyridyl methyl ketone (**45**)], isolated from the African mango (*I. gabonensis*) seeds showed hydroxyl radical scavenging activity with $ED_{50}$ values of 16.7, 11.9, and >20 µM, respectively, vs. quercetin ($ED_{50}$ value: 1.3 µM) [31]. Atawodi [30] used hypoxanthine/xanthine oxidase and 2-deoxyguanosine assays as models to assess the antioxidant activity ($IC_{50}$ value: 28 µL) and radical scavenging capacity ($IC_{50}$ value: 281 µL), respectively, of the methanol extract of *Irvingia gabonensis* seeds [30]. Moreover, Ewere et al. [5] described the DPPH scavenging power of the ethanol extract of *Irvingia gabonensis* leaves with inhibition percentages ranging from 40 to 95% at 20 to 100 µg/mL [5]. FRAP, DPPH, and ABTS tests were used to assess the antioxidant efficacy of the phenol-rich fraction of *Irvingia gabonensis* stem bark with $IC_{50}$ values of 113.10, 18.98, and 18.25 µg/mL, respectively [27].

The antioxidant efficacy of the ethanol extract of *Irvingia gabonensis* seeds was also confirmed by Olorundare et al. [15] by using DPPH, FRAP, and NO scavenging tests. At 25, 50, 75, and 100 µg/mL, this extract inhibited DPPH radicals by 14.59, 43.53, 67.98, and 75.44%, respectively, vs. vitamin C (percent inhibition: 45.06, 56.55, 76.92, and 89.83%, respectively). In FRAP (extract: 0.08, 0.13, 0.28, and 0.48%, vs. vitamin C: 0.24, 0.38, 0.48, and 0.63% at 25, 50, 75, and 100 µg/mL, respectively) and NO (13.55, 39.98, 68.39, and 77.09%, vs. vitamin C: 47.89, 63.09 76.07, and 84.91%, respectively) tests, the plant extract also displayed significant antioxidant activity [15].

To confirm the antioxidant capacity of methanol, ethanol, and phenol-rich extracts from *Irvingia gabonensis* seeds, Nguyen et al. [52] evaluated ferric ion reducing antioxidant power (FRAP), Trolox equivalent antioxidant capacity (TEAC), and 2,2-diphenyl-1-picrylhydrazyl (DPPH) radical scavenging activity and found that these extracts exhibited significant ferric ion reducing power (FRAP values: 0.18, 0.18, and 0.09 mM $Fe^{2+}$/g for the extracts, respectively), TEAC (3597.11, 3046.60, and 21.38 mM Trolox/g for the extracts, respectively), and DPPH radical scavenging potential ($EC_{50}$: 2.81 and 2.81 mg/mL for the methanol and ethanol extracts, respectively). More recently, the *in vitro* and *in vivo* antioxidant activity of the ethanol extract of the *Irvinga gabonensis* leaves was determined by Ewere et al. [53] (Table 7). In the in vitro tests, the extract presented $IC_{50}$ values of 258.47 and 640.05 µg/mL for nitric oxide and hydrogen peroxide scavenging activities, respectively, vs. ascorbic acid ($IC_{50}$: 91.95 and 109.72 µg/mL, respectively). Upon *in vivo* studies, the oral administration of the ethanol extract of *Irvingia gabonensis* leaves to arsenic (dose: 4.1 mg/kg)-induced hepatic oxidative stress rats decreased the serum ALT, ALP, CAT, MDA, and GGT activities and increased the SOD and GPx concentrations, suggesting that this extract possesses antioxidant activity [53]. Furthermore, Atanu et al. [16] confirmed the antioxidant activity of aqueous, ethanol, chloroform, and *n*-hexane extracts from *Irvingia gabonensis* leaves

through DPPH (IC$_{50}$: 30.74, 21.42, 36.62, and 31.41 µg/mL, respectively), FRAP (23.91, 22.25, 22.43, and 11.57 mM Fe$^{2+}$ equivalent, vs. gallic acid: 28.08 mM Fe$^{2+}$ equivalent), and hydroxyl radical (percentage of radical scavenging: 23.02, 81.43, 69.66 and 23.77%, respectively, vs. gallic acid: 100%) inhibition assays [16] (Table 7).

The presence of several classes of compounds such as flavonoids, phenolic compounds, and terpenoids contributes to the antioxidant activity of medicinal plants including *Irvingia* plants. The ability of most of these compounds to interact with reactive oxygen species (ROS) by scavenging or reducing them can characterize their mechanism of action. Controlling the rates of formation and the removal of ROS is a dually essential function. In fact, the intracellular levels of ROS required to perform biological functions should be secured and exceeding ROS levels that reach cytotoxic concentrations should be prevented. For example, flavonoids can exert antioxidant activity by direct radical scavenging and by interacting with the activity of a number of enzymes including nitric oxide synthase and xanthine oxidase [54]. It has also been reported that some flavonoids (which are converted into pro-oxidants during the oxidation process) exert their antioxidant activity indirectly through Nrf2 activation [55,56]. The presence of the benzene ring and the number and position of OH groups in phenolic compounds increase their antioxidant roles. Upon reaction with free radicals, the benzene ring stabilizes the antioxidant compounds. Moreover, phenolics contain H-atoms that can be transferred to free radical substrates to turn them into non-radical substrates (RH, ROH, or ROOH). The latter can contribute to the transfer of electrons or the transition-metal-mediated chelation [57]. The antioxidant mechanism of action has been attributed to three factors including hydrogen transfer, quenching of singlet oxygen, or/and electron transfer [58].

**Table 7.** Antioxidant effects of the extracts and compounds from plants of the genus *Irvingia*.

| *Irvingia* spp. | Extracts/ Compounds | Model of the Study | Significant Results | Toxicity/ Cytotoxicity | Type of Screening | Reference |
|---|---|---|---|---|---|---|
| *Irvingia gabonensis* | Hexane extract of the seeds. | *In vitro* free radical scavenging tests (ORAC, and TEAC). | -FRAP: 283.91 mg/g; -Total antioxidant capacity: 431.58 mg/g. | NS | Oxygen radical antioxidant capacity (ORAC) test/Trolox equivalent antioxidant capacity (TEAC) test. | [50] |
| *Irvingia gabonensis* | Methanol extract of the seeds. | *In vitro* DPPH free radical inhibition. | IC$_{50}$: 177.22 μg/mL. | NS | Colorimetric method. | [51] |
| *Irvingia gabonensis* | Methyl 2-[2-formyl-5-(hydroxymethyl)-1H-pyrrol-1-yl]-propanoate (**39**), 4-[formyl-5-(methoxymethyl)-1H-pyrrol-1-yl]butanoic acid (**40**), and 5-hydroxy-2-pyridyl methyl ketone (**45**). | *In vitro* test. | ED$_{50}$: 16.7, 11.9, >20 μM for compounds **39**, **40**, and **45**, respectively vs. quercetin (1.3 μM). | NS | Fluorometric method using 2′,7′-dichlorofluorescin diacetate. | [31] |
| *Irvingia gabonensis* Baill. | Methanol extract of the seeds. | *In vitro*; hypoxanthine/xanthine oxidase and 2-deoxyguanosine assays as models. | IC$_{50}$: 28 and 281 μL in hypoxanthine/xanthine oxidase assay and 2-deoxyguanosnine tests, respectively. | NS | Hypoxanthine/xanthine oxidase assay and 2-deoxyguanosine assay models. | [30] |
| *Irvingia gabonensis* | Ethanol extract of the leaves. | -*In vitro* (nitric oxide and hydrogen peroxide scavenging properties); -*In vivo*; modulation of the arsenic-induced hepatic oxidative stress in albinos Wistar rats. | -Nitric oxide test: IC$_{50}$: 258.47 μg/mL vs. ascorbic acid (91.95 μg/mL); -Hydrogen peroxide scavenging: IC$_{50}$: 640.05 μg/mL vs. ascorbic acid (109.72 μg/mL); -*In vivo* assay: decrease in serum ALT, ALP, and GGT activities, CAT and MDA concentrations; increase in SOD and GPx concentrations | NS | -Colorimetric methods and enzyme-linked immunosorbent assays; -TECO diagnostic assay kits (Anahaema, CA, USA). | [53] |
| *Irvingia gabonensis* | Phenolic-rich extract from the stem bark. | *In vitro*: anti-oxidant activity (FRAP, DPPH, ABTS tests). | IC$_{50}$ (μg/mL): 113.10, 18.98, and 18.25 in FRAP, DPPH, and ABTS assays, respectively. | NS | Colorimetric methods. | [27] |

**Table 7.** *Cont.*

| *Irvingia* spp. | Extracts/ Compounds | Model of the Study | Significant Results | Toxicity/ Cytotoxicity | Type of Screening | Reference |
|---|---|---|---|---|---|---|
| *Irvingia gabonensis* | Methanol and ethanol extracts and phenol-rich extract from the seeds. | *In vitro* antioxidant assays. | -FRAP: 0.18, 0.18, and 0.09 mM $Fe^{2+}$/g, for the extracts, respectively; <br> -TEAC: 3597.11, 3046.60, and 21.38 mM Trolox/g), for the extracts, respectively; <br> -DPPH: $EC_{50}$: 2.81, 2.81 mg/mL, for methanol and ethanol extracts, respectively. | NS | Colorimetric methods FRAP, TEAC, and DPPH assays. | [52] |
| *Irvingia gabonensis* | Aqueous, ethanol, chloroform, and n-hexane extracts from the leaves. | *In vitro* antioxidant tests (DPPH, FRAP, $OH^-$). | -DPPH: $IC_{50}$: 30.74, 21.42, 36.62, and 31.41 µg/mL for the extracts, respectively, vs. butylated hydroxytoluene (21.73 µg/mL); <br> -FRAP: 23.91, 22.25, 22.43, and 11.57 mM Fe2+ equivalent for the extracts, respectively, vs. gallic acid: 28.08 mM Fe2+ equivalent; <br> -Percentage of inhibition of $OH^-$ radicals: 23.02, 81.43, 69.66, and 23.77% for the extracts, respectively, vs. gallic acid (100%). | NS | Colorimetric methods for DPPH, FRAP and $OH^-$ radical scavenging tests, respectively. | [16] |

ABTS: 2,2′-azino-bis(3-ethylbenzothiazoline-6-sulfonic acid); ALP: alkaline phosphatase; ALT: alanine aminotransferase; CAT: catalase; DPPH: 2,2-diphenyl-1-picrylhydrazyl; FRAP: ferric reducing antioxidant power; GGT: gamma-glutamyl transferase; GPx: glutathione peroxidase; $IC_{50}$: half maximal inhibitory concentration; MDA: malondialdehyde; NS: not specified; ORAC: oxygen radical absorbance capacity; SOD: superoxide dismutase; TEAC: Trolox equivalent antioxidant capacity.

### 4.4. Antidiabetic Activity

To evaluate the antidiabetic activity of aqueous, ethanol, chloroform, and *n*-hexane extracts from *Irvingia gabonensis* leaves, Atanu et al. [16] performed α-amylase and α-glucosidase inhibition tests. In the α-amylase test, the aqueous, ethanol, chloroform, and *n*-hexane extracts inhibited the tested enzyme with $IC_{50}$ values of 30, 45, 130, and 75 μg/mL, respectively, vs. acarbose ($IC_{50}$ value: 55 μg/mL). Moreover, these extracts afforded $IC_{50}$ values of 10, 15, 18, and 60 μg/mL (vs acarbose: 35 μg/mL), respectively, in the α-glucosidase inhibition test. In another study, the antidiabetic activity of terminalin (**32**), a phenolic compound isolated from the aqueous extract of *Irvingia gabonensis* seeds, was evaluated through the inhibition of protein tyrosine phosphate enzymes (PTPs). Terminalin (**32**) demonstrated antidiabetic activity via *in vitro* inhibition (80%) of the catalytic sites of PTPN1, PTPN9, PTPN11, and PTPRS, which hampered the glucose uptake in differentiated C2C12 myocytes [29].

### 4.5. Other Biological Activities

In an *in vitro* study, Chung et al. [59] demonstrated the inhibitory effect of the methanol leaf extract of *Irvingia malayana* on the human receptor 5HT1a with an inhibition rate of 65%, inferring that this extract can overcome disorders of the central nervous system such as migraine, sleeping disorders, Alzheimer's disease, epilepsy, and Parkinson's disease [59]. Two fractions prepared from the methanol extract of *Irvingia malayana* inhibited the growth of the human leukemia cell-line HL60 by 73.9% and 46.3%, respectively [60]. Ng et al. [25] used the rat aortic ring assay to screen the antiangiogenic activity of the methanol extract of *Irvingia malayana* and its isolate betulinic acid. At 100 μg/mL, both the extract and betulinic acid inhibited the human umbilical vein endothelial cells by 46.5 and 45.5%, respectively, vs. suramin (percent inhibition: 55.5%). Furthermore, the studied extract revealed no cytotoxic activity against HepG2, HCT-116, T-47D, NCI-H23, and CCD-18Co cells, as the $IC_{50}$ values were 88.23, 54.94, 64.32, 78.12, and 111.34 μg/mL, respectively [25]. In a randomized clinical trial, Méndez-Del Villar et al. [61] described the effect of *I. gabonensis* administration in reverting metabolic syndrome (seven patients out of 12 patients; 58.3%) compared with the group of patients who received placebo (two patients remitted out of 12 patients; 16.7%). These results attested to the effectiveness of *Irvingia* against metabolic diseases such as diabetes [61]. The ethanol extract of *I. gabonensis* leaves inhibited the growth of the worm *Heligmosomoides bakeri* with percent inhibition values of 71.43, 57.14, and 42.9% at concentrations of 500, 250, and 125 mg/mL, respectively, suggesting that the *I. gabonensis* extract possesses anthelminthic activity [62]. To evaluate the protective role of *Irvingia gabonensis*, the ethanol extract of *I. gabonensis* leaves (at 250 and 500 mg/kg) was administered to male Wistar rats with sodium arsenite (2.5 mg/kg)-induced toxicity. This extract decreased the activities of ALT (52.71, 57.24, 40.72, and 39.65 U/L), AST (9, 9.46, 9.23, and 8.92 U/L), and γGT (5.21, 3.47, 6.94, and 4.63 U/L), respectively, when compared to the group treated only with sodium arsenite [ALT (78.61 U/L), AST (22.99 U/L), and γGT (10.42 U/L)] [4]. Recently, Ugwu et al. [63] used carrageenan-induced edema to evaluate the anti-inflammatory activity of solid lipid microparticles prepared from unPEGylated lipid matrices of the *Irvingia* fat matrix. *Irvingia*-loaded microparticles significantly reduced the volume of edema in rats by 38, 40, 87, 65, and 67% after 0.5, 1, 2, 3, and 4 h, respectively [63] (Table 8).

**Table 8.** Other pharmacological activities of the extracts and compounds from *Irvingia* plants.

| *Irvingia* spp. | Extracts/ Compounds | Model of the Study | Significant Results | Toxicity/ Cytotoxicity | Type of Screening | Reference |
|---|---|---|---|---|---|---|
| *Irvingia gabonensis* | Water and ethanol extract prepared from the stem bark. | Male mice (*in vivo*). | -Reduction of the locomotion in mice treated with water extract (250–750 mg/kg); <br> -Production of time- and dose-related analgesia by both extracts (250–750 mg/kg). | NS | Hot plate test. | [64] |
| *Irvingia gabonensis* | Aqueous extract of leaves. | -Gastrointestinal motility test in mice; <br> -Castor oil-induced diarrhea in mice (*in vivo*). | -Decrease in the gastrointestinal motility: 40.12, 39.45 and 37.45% at the doses of 100, 200 and 400 mg/kg, respectively; <br> -Protection of mice by 71.43, 81.63, 83.27% at 100, 200 and 400 mg/kg of extract, respectively. | NS | -Gastrointestinal motility test in mice; <br> -Castor oil-induced diarrhea in mice. | [11] |
| *Irvingia gabonensis* | Ethanol (80%) extract prepared from the leaves. | Model: The worm *Heligmosomoides bakeri*. | Percentage of larvae death: 71.43, 57.14, and 42.9% of larval deaths at the concentrations of 500, 250, and 125 mg/mL, respectively. | NS | Cell viability. | [62] |
| *Irvingia gabonensis* | Ethanol (50%) extract from the leaves. | Sodium arsenite-induced hepatotoxicity in male Wistar albino rats (*in vivo*). | Decrease in the activity of serum biochemical parameters: <br> -ALT: 52.71, 57.24, 40.72, and 39.65 U/L; respectively; <br> -AST: 9, 9.46, 9.23, and 8.92 U/L; respectively; <br> -$\gamma$GT: 5.21, 3.47, 6.94, and 4.63 U/L, respectively, when compared with the group treated with sodium arsenite alone [ALT (78.61 U/L), AST (22.99 U/L), and $\gamma$GT (10.42 U/L)]. | NS | Sodium arsenite-induced hepatotoxicity in male Wistar albino rats. | [4] |
| *Irvingia gabonensis* | Aqueous, methanol and hexane extracts from the leaves. | *In vivo* anti-diarrheal activity in rat models. | Percentage of protection: <br> -Aqueous extract: 80% at 100 and 200 mg/kg, vs. loperamide (80% at 2 mg/kg); <br> -Methanol extract: 80% protection at 200 mg/kg, vs. loperamide (80% at 2 mg/kg); <br> -Hexane extract: 40 and 60% protection at 100 and 200 mg/kg, respectively. | NS | NS | [43] |
| *Irvingia wombolu* | Solid lipid microparticles prepared from unPEGylated lipid matrices of *Irvingia* fat matrix (*Irvingia*-loaded microparticles). | Rat paw edema model (*in vivo*) | Reduction of the volume of edema in rats in percentages: 38, 40, 87, 65, and 67% after 0.5, 1, 2, 3, and 4 h, respectively. | NS | Carrageenan rat paw edema test. | [63] |
| *Irvingia gabonensis* | Aqueous, ethanol, chloroform, and *n*-hexane extracts from the leaves. | *In vitro* antidiabetic test using $\alpha$-amylase and $\alpha$-glucosidase inhibition tests | -$\alpha$-amylase test: <br> IC$_{50}$: 30, 45, 130, and 75 $\mu$g/mL, respectively, vs. acarbose (IC$_{50}$: 55 $\mu$g/mL); <br> -$\alpha$-glucosidase inhibition test: <br> IC$_{50}$: 10, 15, 18, and 60 $\mu$g/mL, respectively (vs. acarbose: 35 $\mu$g/mL) | NS | $\alpha$-Amylase and $\alpha$-glucosidase inhibition tests. | [16] |

ALT: alanine aminotransferase, AST: aspartate aminotransferase, $\gamma$GT: gamma-glutamyl transferase; IC$_{50}$: half maximal inhibitory concentration, NS: not specified, PEG: polyethylene glycol.

## 5. Toxicity Profile of *Irvingia* spp.

A number of studies have reported on the toxicity profile of plants from the genus *Irvingia*. For example, in a study published by Kothari et al. [65], the oral administration of an *Irvingia gabonensis* extract at 0, 100, 1000, and 2500 mg/kg for 90 days did not induce any clinical signs or mortality in Sprague Dawley rats. Accordingly, the NOAEL (no observed adverse effect level) for the *I. gabonensis* extract was predicted to be more than the highest dose tested (i.e., 2500 mg/kg bw/day (LD$_{50}$)).

Clinical symptoms, body weight, feed consumption, and mortality were not observed in the treated rats. Moreover, there were no significant changes in the hematological and biochemical parameters of rats [65]. In another study, the oral administration of aqueous extracts from *I. gabonensis* stem bark and leaves to male Wistar rats did not cause significant changes compared to the negative control (without treatment). In brief, the oral treatment of rats with three doses (100, 1000, and 2000 mg/kg) of an aqueous leaf extract of *I. gabonensis* did not affect the serum biochemical markers of toxicity such as ALT (63.30, 59.50, and 58.00 IU/L at 100, 1000, and 2000 mg/kg, respectively, vs. the untreated control: 66.50 IU/L), AST (158.33, 159.33, and 158.00 IU/L at 100, 1000, and 2000 mg/kg, respectively, vs. the untreated control, 159.16 IU/L) and ALP (127.90, 102.00, and 121.13 IU/L at 100, 1000, and 2000 mg/kg, respectively, vs. the untreated control, 103.50 IU/L), suggesting that these extracts might not affect the liver physiology [66]. In this study, the authors confirmed the NOAEL (2500 mg/kg) reported previously by Kothari et al. [65].

To evaluate the potential of an ethanol extract from *Irvingia gabonensis* seeds in reverting the cardiotoxicity caused by doxorubicin in mice, Olorundare et al. [15] administered 100, 200, and 400 mg/kg to doxorubicin (15 mg/kg)-mediated cardiotoxicity and determined the level of serum cardiac enzymes such as cardiac troponin I (cTI) and lactate dehydrogenase (LDH) as well as cardiac tissue oxidative stress markers [catalase (CAT), malonyldialdehyde (MDA), superoxide dismutase (SOD), glutathione-S-transferase (GST), glutathione peroxidase (GSH-Px), and reduced glutathione (GSH)]. The plant extract increased the levels of serum GSH [21.3, 17.0, 19.6, and 24.7 U/L at 20, 100, 200, and 400 mg/kg, respectively, vs. the Dox-treated group (14.8 U/L) and negative control (18.8 U/L)], GST [2.6, 2.0, 2.4, and 3.0 U/L at 20, 100, 200, and 400 mg/kg, respectively, vs. the doxorubicin -treated group (1.1 U/L) and negative control (1.6 U/L)], GPx [2.4, 1.3, 2.3, and 3.3 U/L at 20, 100, 200, 400 mg/kg, respectively, vs. the doxorubicin-treated group (1.0 U/L) and negative control (1.2 U/L)], SOD [11.5, 11.1, 12.8, and 15.4 at 20, 100, 200, and 400 mg/kg, respectively, vs. the doxorubicin-treated group (6.9 U/L) and negative control (9.5 U/L)], and CAT [51.4, 54.2, 56.6, and 78.7 U/L at 20, 100, 200, and 400 mg/kg, respectively vs. doxorubicin-treated group (16.7 U/L) and negative control (43.6 U/L)]. In contrast, there was a decrease in the levels of MDA [5.2, 5.3, 3.9, and 3.5 U/L at 20, 100, 200, and 400 mg/kg, respectively, vs. the doxorubicin-treated group (12.8 U/L) and negative control (4.3 U/L)] [15].

## 6. Traditional Uses of *Irvingia* spp.

The genus *Irvingia* encompasses seven species, among which six occur in tropical Africa and one in Southeast Asia.

Seven species of *Irvingia* are abundant in the African continent and include *Irvingia gabonensis* (syn. *I. pauciflora* Tiegh., *I. platycarpa* Tiegh., and *I. tenuifolia* Hook), *I. robur*, *I. smithii*, *I. excels*, *I. grandiofolia*, *I. wombolu* (syn. *I. tenuinucleata*), and *I. malayana* (syn. *I. oliveri*). Among these, *I. gabonensis* and *I. wombulu* contain sweet and bitter comestible pulps, respectively, which are used as traditional foods by many people, especially in the countryside [3]. The seeds of *I. gabonensis* are commercially available and are marketed across a number of African countries including Liberia, Nigeria, Sierra Leone, Cameroon, Gabon, and Côte d'Ivoire [3]. These plant species are used in traditional medicine to treat a number of disease conditions including diarrhea, scabies, toothache, yellow fever, inflammation, and liver and gastrointestinal disorders [53] (Table 9).

In Cameroon, the decoction of *I. gabonensis* and *I. wombolu* seeds is administered twice a day until body weight is reduced [8,67]. In Gabon, the bark decoction of *I. grandifolia* (common name: Mulenda) is used in a bath to treat asthenia; however, in children, the leaves are pounded and mixed with food to overcome this disease condition [68]. Commonly called Aslotin in Benin, *Irvingia gabonensis* is used to treat icterus [69]. In Nigeria, *Irvingia tenuinucleata* is also called Bush mango or Oro, and its fruits and bark have been used traditionally to treat diarrhea, scabies, toothache, yellow fever, inflammation, and diabetes. *Irvingia gabonensis* has also been indicated for the treatment of dysentery in this country [10]. In Laos, the bark or wood is grilled on fire, boiled in salty water, and drunk to obtain relief from various diseases [9,41] (Table 9).

**Table 9.** Traditional uses of *Irvingia* spp.

| Plant Species | Common Names | Plant Organs Used/ Route of Administration | Traditional Uses | Country of Use | References |
|---|---|---|---|---|---|
| *Irvingia gabonensis* (Aubry-Lecomte ex O'Rorke) Baill. | Aslotin | Leaves | Used to treat icterus. | Benin | [69] |
| -*Irvingia gabonensis* (Aubry-Lecomte ex O'Rorke) Baill.; -*Irvingia wombolu* Vermoesen (Syn: *Irvingia tenuinucleata* Tiegh.). | bojep | Seed/oral administration | The seeds are used as condiments and the decoction is administered (a bowlful 2 times a day until recovery) to reduce body weight. | Cameroon | [8,67] |
| *Irvingia grandtfolia* (Engl.) Engl | Mulenda | Bark/leaves | The bark decoction is used in a bath to treat asthenia. For children, the leaves are pounded and mixed with food. | Gabon | [68] |
| *Irvingia malayana* Oliver ex Bennett | Bohk; Yarr/Niharr | Bark/wood | The bark or wood is grilled on fire and boiled with salt in water and administered orally to overcome tuberculosis related symptoms. | Laos | [9,41] |
| -*Irvingia gabonensis* (Aubry-Lecomte ex O'Rorke) Baill.; -*Irvingia tenuinucleata* Tiegh. *Irvingiaceae.* | -Bush mango, Oro, dika nut tree, Ugiri, Goron, Biri, and Apon -Bush mango, Oro | -Fruit, bark, seeds and roots -Fruit and bark | -These organs are used as food and to get relief from dysentery, diarrhea toothache and diabetes; -These organs are used to relieve dysentery, diarrhea toothache and diabetes. | Nigeria | [10] |

### 7. Critical Assessment and Discussion

The present work discusses recent developments in the traditional uses, phytochemical composition, and pharmacological and toxicological studies of plants from the genus *Irvingia*. Research gaps that have not been explored thus far are also presented and discussed.

At least six species (*Irvingia gabonensis*, *I. grandifolia*, *I. tenuinucleata*, *I. malayana*, *I. wombolu*, and *I. midbr*, among others) of *Irvingia* are traditionally used to relieve a number of diseases including diarrhea, scabies, toothache, yellow fever, inflammation, liver and gastrointestinal disorders, and others. In traditional medicine, the fruit, bark, leaves, seeds, and roots are the most commonly used organs of *Irvingia* plants. Decoction is the main mode of preparation, while the main mode of administration is via the oral route. Noteworthy, *Irvingia gabonensis* has been the most widely studied species of the genus *Irvingia*. Except for betulinic acid (**2**), which was also pinpointed to originate from *Irvingia malayana*, the remaining compounds were identified from various organs [stem bark (compounds **1**–**4**, **13**, **14**, [23]), seeds (compounds **15**, **16**, and **33** [30]; compounds **6**–**9**, **13**, **16**–**30**, and **33**–**38** [26]; compounds **31**, **39**–**45**, [31]) of *Irvingia gabonensis*. Most importantly, these compounds were found to exhibit antimicrobial (compounds **1**–**4**, **13**, and **14** [23]) and antioxidant (compounds **39**, **40**, and **45** [31]) activities. Nevertheless, a substantial amount of extracts were prepared from *Irvingia* plants and evaluated for a number of biological activities including antiprotozoal, antimicrobial, antioxidant, antidiabetic, antiangiogenic, and anthelmintic activities.

In antiprotozoal assays, three plant species were involved including *Irvingia gabonensis* (antiplasmodial and anti-*Trypanosoma brucei*; $IC_{50}$: <8 μg/mL), *Irvingia malayana* (antiplasmodial) and *Irvingia grandifolia* (antileishmanial activity; $IC_{50}$ values: <8.4 μg/mL), whereas the solvents mostly used for extraction comprised ethanol, methanol, and dichloromethane. Stem bark and leaves were the plant organs tested. As already discussed, the seeds of *Irvingia* spp. contain a number of secondary metabolites (compounds **6**–**9**, **17**–**31**, **33**–**45**) that can be further isolated and characterized as antiprotozoal compounds. In most of the reported studies, the cytotoxicity of the plant extracts was not investigated against human mammalian cells. The antiprotozoal mechanisms of action of the extracts and compounds from *Irvingia* spp. are still unexplored. Thus, research breaches to be filled in antiprotozoal drug screening include the following: (i) antiprotozoal guided-fractionation of the constituents from all known and available species (>5) of *Irvingia* should be performed; (ii) antiprotozoal experiments should be followed by cytotoxicity assays to depict the selectivity of test samples; (iii) the polarity of solvent used for extraction should be varied to extract both polar and nonpolar potential antiprotozoal compounds from *Irvingia* spp.; (iv) appropriate negative and positive controls should be considered in antiprotozoal assays; (v) in-depth toxicity studies and antiprotozoal mechanisms of action of the most promising compounds should be explored. In antimicrobial assays, numerous reports related to the screening of plant extracts are widespread across the literature; however, only a few authors [23] have isolated and characterized antimicrobial compounds from *Irvingia gabonensis*. Therefore, it will be of interest for researchers working on antimicrobial drug discovery, to (i) explore other *Irvingia* species and their constituents with respect to antimicrobial drug screening, (ii) screen extracts and compounds against dermatophytes (Epidermophyton, *Microsporum*, and *Trichophyton* spp.) and associated fungi that cause superficial infections of the skin, hair and nails, and (iii) perform toxicity studies and antimicrobial mechanisms of action of the most promising compounds.

Although several methods (FRAP, ABTS, DPPH, and NO scavenging tests, etc.) have been used to screen extracts and compounds from *Irvingia* plants, *Irvingia gabonensis* was the only plant species mostly involved in biological tests. In fact, the ability of DPPH and ABTS to eliminate hydroxyl and superoxide radicals is attributable to their high hydrogen-donating capacity [70]. Specifically, the ABTS test is based on the production of a blue/green ABTS•+, which is reduced by antioxidants; whereas the DPPH assay is established on the reduction of the purple DPPH• to 1,1-diphenyl-2-picryl hydrazine. In contrast, the FRAP test differs from the DPPH and ABT assays as there are no free radicals

involved, but the reduction of ferric iron ($Fe^{3+}$) to ferrous iron ($Fe^{2+}$) by antioxidants has been examined [71]. Although there was an adequate correlation with results obtained from the antioxidant efficacy of *Irvingia gabonensis* extracts using the FRAP and DPPH methods [15,27,52], the presence of polyphenols in plants, in general, has explained the antioxidant nature of plants [72,73], and the use of several approaches to study the anti-radical scavenging activity might influence the outcome of the tests. On the other hand, studying the antioxidant activity of plants and their secondary metabolites might help to better elucidate the mechanisms of action of bioactive extracts or compounds. In the case of *Irvingia*, a study of the antioxidant activity of other *Irvingia* species (such as *Irvingia malayana* and *Irvingia grandifolia*) and compounds thereof is warranted. A couple of antidiabetic experiments were recently conducted by Atanu et al. [16] and Yoon et al. [29] using enzyme inhibition assays (inhibition of α-amylase, α-glucosidase, and protein tyrosine phosphatase). However, these experiments provide an idea about the *in vitro* antidiabetic effect of only one species of *Irvingia* (i.e., *Irvingia gabonensis*). Extending antidiabetic studies to other *Irvingia* species might afford more insights into the antidiabetic potential of *Irvingia* spp. and their secondary metabolites. Other biological tests on the extracts and compounds from *Irvingia* spp. included antiangiogenic (isolate of *I. malayana*: betulinic acid; [25]), anthelmintic (ethanol extract of *I. gabonensis* leaves; *Heligmosomoides bakeri*; percent inhibition: 42.9% at 125 mg/mL; [62], and hepatoprotective (sodium arsenite-induced toxicity in male Wistar rats; decrease in ALT and AST enzymes; *I. gabonensis*; [4]) activities. However, there is almost no information on the secondary metabolites (active principles) that are responsible for the observed biological activities. As *Irvingia gabonensis* is the most investigated plant of the genus *Irvingia*, phytochemical and biological studies of the other species (>5) of *Irvingia* are warranted. Overall, the genus *Irvingia* comprises at least six species (*Irvingia gabonensis*, *I. grandifolia*, *I. tenuinucleata*, *I. malayana*, *I. wombolu,* and *I. midbr*) with ethnomedicinal indications. Various organs (leaves, stem bark, seeds, among others) of these species are used traditionally to overcome a number of diseases such as diarrhea, yellow fever, scabies, toothache, inflammation, dysentery as well as liver and gastrointestinal disorders.

Modern pharmacological studies have revealed antiprotozoal, antimicrobial, antidiabetic, antioxidant, anti-inflammatory, and hepatoprotective activities, thus validating the ethnomedicinal uses of *Irvingia* plants in the treatment of fever and inflammation conditions and bacterial gastroenteritis (vomiting, diarrhea). Furthermore, several studies have revealed the presence of more than forty (40) compounds from *Irvingia* plants, mostly phenolic compounds, flavonoids, and terpenoids. These compounds might be responsible for the reported biological activities of *Irvingia* plants. Nevertheless, there are research gaps that need to be filled regarding the potential application of *Irvingia* spp., from their ethnopharmacological validation and phytochemical evaluation to preclinical and clinical investigations. These include: (i) the lack of *in vitro* and *in vivo* toxicity tests and mechanistic studies on the extracts and compounds from *Irvingia* spp. in reported works, (ii) a lack of appropriate controls in most of the experiments, (iii) only a few of the existing species of *Irvingia* have been studied up until now and most of the traditional indications (jaundice and related symptoms, scabies, asthenia, or respiratory ailments, and so on) have not been scrutinized using modern pharmacological experiments; (iv) compared to *Irvingia* crude extracts, less information is available regarding biological studies of pure compounds from *Irvingia*. Therefore, (a) more studies on *in vitro* and *in vivo* toxicity tests as well as mechanistic studies of the extracts and compounds from *Irvingia* spp. should be investigated; (b) robust and comprehensive phytochemical analyses such as LC-MS/MS, GC/MS, HPTLC, and UHPLC-ESI-Q-TOF-MS, in association with biological screening (bioassay-guided), should be performed to substantially identify the bioactive compounds from *Irvingia* species; (c) other *Irvingia* species and traditional indications should be considered in modern pharmacological studies. These studies might contribute to the potential therapeutic applications of *Irvingia* in the treatment of various diseases.

## 8. Conclusions and Perspectives

The present study provides a comprehensive analysis regarding the traditional uses, phytochemical analyses, and pharmacological and toxicity studies of plants from the genus *Irvingia*. *Irvingia* plants are used for the traditional treatment of a number of diseases such as diarrhea, yellow fever, scabies, toothache, inflammation, dysentery, and liver and gastrointestinal disorders. Previous reports have revealed a number of pharmacological activities including antiprotozoal, antimicrobial, antidiabetic, antioxidant, anti-inflammatory, and hepatoprotective activities. The reported activities have been attributed to the existence of a number of secondary metabolites (flavonoids, phenolic compounds, terpenoids, tannins, saponins, alkaloids, etc.) in *Irvingia* plants. On the other hand, *Irvingia* species such as *Irvingia gabonensis*, which is used as a food and as a weight loss supplement, can be a potential remedy in treating cardiovascular diseases. Nevertheless, a number of studies (*in vitro* and *in vivo* toxicity and in-depth mechanistic experiments, clinical trials, additional phytochemical and pharmacological studies of the uninvestigated species of *Irvingia*) are needed for the successful utilization of *Irvingia* plants in the treatment of various diseases.

**Author Contributions:** Conceptualization, B.P.K. and F.F.B.; Methodology, B.-N.N.-D. and B.P.K.; Software, B.P.K.; Validation, B.P.K.; Formal analysis, B.-N.N.-D. and B.P.K.; Investigation, B.-N.N.-D. and B.P.K.; Resources, B.P.K.; Data curation, B.-N.N.-D., B.P.K. and P.K.L.; Writing—original draft preparation, B.-N.N.-D. and B.P.K.; Writing—review and editing, B.-N.N.-D., B.P.K. and P.K.L.; Visualization, B.P.K.; Supervision, F.F.B. and B.P.K.; Project administration, F.F.B. and B.P.K.; Funding acquisition, F.F.B. and B.P.K. All authors have read and agreed to the published version of the manuscript.

**Funding:** This research received no external funding.

**Institutional Review Board Statement:** Not applicable.

**Informed Consent Statement:** Not applicable.

**Conflicts of Interest:** The authors declare no conflict of interest.

## Abbreviations

ABTS: 2,2'-azino-bis(3-ethylbenzothiazoline-6-sulfonic acid); ADP: Arthemether-Dika (*Irvingia*) fat-Phospholipon; ALP: alkaline phosphatase; ALT: alanine aminotransferase; AST: aspartate aminotransferase; CAT: catalase; cTI: cardiac troponin I; DNA: deoxyribonucleic acid; DPPH: 2,2-diphenyl-1-picrylhydrazyl; $ED_{50}$: median effective dose; FRAP: ferric reducing antioxidant power; GC/MS: gas chromatography–mass spectrometry; GGT: gamma-glutamyl transferase; GPx: glutathione peroxidase; GSH: reduced glutathione; GSH-Px: glutathione peroxidase; γGT: gamma-glutamyl transferase; HCT-116: human colorectal carcinoma cell line; HeLa: human cervix carcinoma cells; HepG2: human liver cancer cell line; HPTLC: high-performance thin layer chromatography; $IC_{50}$: half maximal inhibitory concentration; *Irvingia* spp.: *Irvingia* species; LC-MS/MS: liquid chromatography with tandem mass spectrometry; $LD_{50}$: median lethal dose; LDH: lactate dehydrogenase; LM: lipid matrices; MDA: malondialdehyde; MIC: minimum inhibitory concentration; MRC5: human diploid embryonic lung cells; MRSA: Methicillin-resistant *Staphylococcus aureus*; MTT: 3-(4,5-dimethylthiazol-2-yl)-2,5-diphenyltetrazolium bromide; NCI-H23: human non-small cell lung carcinoma; human fibroblast cell line; NO: nitric oxide; NOAEL: no observed adverse effect level; Nrf2: nuclear factor erythroid 2–related factor 2; NS: not specified; NT: not tested; ORAC: oxygen radical absorbance capacity; P90G: Phospholipon 90 G containing about 94.0% of phosphatidylcholine with 0.1% ascorbyl palmitate; PEG: polyethylene glycol; PTPs: protein tyrosine phosphate enzymes; PTPN1: protein tyrosine phosphatase non-receptor type 1; PTPN9: PTPN type 9; PTPN11: PTPN type 11; PTPRS: protein tyrosine phosphatase receptor type S; ROS: reactive oxygen species; SA: *Staphylococcus aureus*; SI: Selectivity index; SOD: superoxide dismutase; T-47D: human epithelial cell line derived from a mammary ductal carcinoma; TEAC: Trolox equivalent antioxidant capacity; UHPLC-ESI-Q-TOF-MS: ultra-high performance liquid chromatography-quadrupole time-of-flight mass spectrometry; UHPLC/HRMS: ultra-high-performance liquid chromatography/high resolution mass spectrometry.

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
