# Peer review of "Ethnomedicinal Uses, Phytochemistry, and Pharmacological Activity of the Irvingia Species"

_ddc, doi:10.3390/ddc2040042_

Round 1

Reviewer 1 Report

Dear Authors,

It's really an interesting review. In order to be published you need to do an extra effort to improve your manuscript.

Here you have some suggestions and questions for enhancing it:

1. Introduction:

Please take into account these update reviews, just as an example: 

https://pubmed.ncbi.nlm.nih.gov/36816319/

https://pubmed.ncbi.nlm.nih.gov/31793087/

https://pubmed.ncbi.nlm.nih.gov/?term=Irvingia&filter=pubt.review

2. Research methods:

However you made a wide search you need to include the in out criteria. For better understanding of your results of the  literature search, you may add a flow chart showing the items found in every databases, the excluded and included items and the final results analysed.

3. Traditional uses:

As the main objective of Table 1 is to show traditional uses of different species of Irvingia, you should organize the info begining with species, common name, part used, traditional uses and preparations and finally country of use. If you want to provide the info as you presented you should underline traditionals uses and species. 

4. Phytochemistry:

In order to display the different phytocompounds of Irvingia in Figure 1, they should be presented by groups (saponins, alkaloids, tannins, etc.) and there is not necessary to be large chemical structures.

With regards to the Figure 2, it could be useful to include type of preparation together with plant organs.

What kind of category is OSE? Ellagitannin and tannins belong to the same category, don't they? Please check how you showed Table 2.

There is not any type of alkaloids, you mentioned in abstracts!!

5. Pharmacological:

Table 3 should summarize, clearly, which are in vitro and in vivo assays. 

In addition, you should add a footnote (full name to the abreviations) to the Table 3, 4, 5 and 6.

Line 311, Table 3 or Table 4???.

You need to check Table 6, in order to differenciate in vitro and in vivo assays and type of activity.

6. Toxicity profile:

You need to mention the DL50 or/and NOAEL for any types of toxicological studies (short or long term), if possible.

7. Critical assessment and discussion:

You need to point out which of the 7 sp. are the most studied? 

8. Conclusion

According to your review, could you recommend the use of Irvingia in some type of diseases?

You need to indicate that Irvingia should be deeply studied not only in vitro and  in vivo but also clinical trials. In effect, there some number of systematic reviews and metanalysis.

I hope you receive this feedback as useful as constructive for improving your work.

Good luck!!!  

Author Response

Manuscript ID: ddc-2604855

To,

The reviewer

Dear Reviewer,

We thank you for the optimistic comments toward our manuscript. We have now revised the manuscript in accordance with the comments that you have raised.

The point by point reply to the comments is as follows:

Dear Authors,

It's really an interesting review. In order to be published you need to do an extra effort to improve your manuscript.

Reply: We thank the reviewer for the positive view toward our manuscript.

Here you have some suggestions and questions for enhancing it:

  1. Introduction:

Please take into account these update reviews, just as an example: 

https://pubmed.ncbi.nlm.nih.gov/36816319/

https://pubmed.ncbi.nlm.nih.gov/31793087/

https://pubmed.ncbi.nlm.nih.gov/?term=Irvingia&filter=pubt.review

Reply: We agree with the reviewer’s inquisitiveness. We have now taken into consideration the work mentioned in the links provided.

  1. Research methods:

However you made a wide search you need to include the in out criteria. For better understanding of your results of the  literature search, you may add a flow chart showing the items found in every databases, the excluded and included items and the final results analysed.

Reply: We agree with the reviewer’s suggestion. A flow chart (Figure 1) has now been incorporated at the appropriate place of the main manuscript.

  1. Traditional uses:

As the main objective of Table 1 is to show traditional uses of different species of Irvingia, you should organize the info begining with species, common name, part used, traditional uses and preparations and finally country of use. If you want to provide the info as you presented you should underline traditionals uses and species. 

Reply: The changes have been amended as suggested.

  1. Phytochemistry:

In order to display the different phytocompounds of Irvingia in Figure 1, they should be presented by groups (saponins, alkaloids, tannins, etc.) and there is not necessary to be large chemical structures.

Reply: We agree with the reviewer’s observation. The chemical compounds from plants of the genus Irvingia have now been grouped into phytochemical classes.

With regards to the Figure 2, it could be useful to include type of preparation together with plant organs.

Reply: The mode of plant preparation has been added to the text.

What kind of category is OSE? Ellagitannin and tannins belong to the same category, don't they? Please check how you showed Table 2.

Reply: Actually, oses are a class of sugars, i mean terminology used for a class of sugars. We have now checked Table 2 and corrections have been implemented as proposed. The changes have been colored red with yellow stripes.

There is not any type of alkaloids, you mentioned in abstracts!!

Reply: Alkaloids are a huge group of naturally occurring organic compounds, which contain nitrogen atoms or atoms (amino or amido in some cases) in their structures. Such compounds are widespread across this manuscript (e.g.: methyl 2-[2-formyl-5-(hydroxymethyl)-1H-pyrrol-1-yl]-propanoate, 4-[formyl-5-(methoxymethyl)-1H-pyrrol-1-yl]butanoic acid, and 5-hydroxy-2-pyridyl methyl ketone, etc.)

  1. Pharmacological:

Table 3 should summarize, clearly, which are in vitro and in vivo assays. 

Reply: The changes have now been amended as recommended.

In addition, you should add a footnote (full name to the abreviations) to the Table 3, 4, 5 and 6.

Reply: We agree with the reviewer’s suggestion. Footnotes have been added to the tables. A separate list of abbreviations has also been provided to the text.

Line 311, Table 3 or Table 4???.

Reply: We agree with the reviewer’s apprehension. The change has been amended as recommended.

You need to check Table 6, in order to differenciate in vitro and in vivo assays and type of activity.

Reply: The changes have been amended as recommended.

  1. Toxicity profile:

You need to mention the DL50 or/and NOAEL for any types of toxicological studies (short or long term), if possible.

Reply: The changes have been amended as proposed. DL50 of Irvinvia samples have been incorporated wherever possible with the manuscript.

  1. Critical assessment and discussion:

You need to point out which of the 7 sp. are the most studied? 

Reply: Actually, Irvingia gabonensis has been the most widely studied species of the genus Irvingia. This statement has now been incorporated in the section “Critical assessment and discussion”.

  1. Conclusion

According to your review, could you recommend the use of Irvingia in some type of diseases?

Reply: We believe that some Irvingia species, such as Irvingia gabonensis, which is used as a food, and as weight loss supplements can be a potential remedy to treating cardiovascular diseases.

You need to indicate that Irvingia should be deeply studied not only in vitro and  in vivo but also clinical trials. In effect, there some number of systematic reviews and metanalysis.

Reply: The details (in vitro, in vivo tests, and clinical trials) have now been incorporated in the conclusion section as suggested.

I hope you receive this feedback as useful as constructive for improving your work.

Reply: We thank the reviewer for the optimistic comments toward our manuscript.

Good luck!!!  

Reply: Thank you.

With kind regards,

Reviewer 2 Report

1.       The logical work arrangement should be changed. First, the chemistry should be described (chapter 4) and then the activity (chapter 3). I would even suggest mentioning the traditional use at the end. The review paper is about scientific proof of the effect of the plant material.

2.       Figure 1 and Table 2 should be combined. Table 2 does not systematize and divide the compounds, e.g. into phenolic acids, flavonoids, etc. in the appropriate order. There is a lot of chaos that makes it difficult to understand the work. I don't understand whether their listing is accidental or whether they were listed in order of the highest content in the plant material?

3.       Chapter 4 lacks information about the content of individual compounds, or even suggestions about which compounds may contain the most.

4.       Many MIC values are given in section 5.2. They make reading very difficult and it's hard for me to understand what they refer to. I suggest changing the form of presenting this data.

5.       In chapter 5.3, the authors refer to the Folin-Ciocalteu method, which assesses the total polyphenol content. This is not a method that provides antioxidant activity. There is a high correlation between TPC and activity, but there was a large factual error in the work.

6.       Are you sure the inhibition of acetylcholinesterase (AChE) and butyrylcholinesterase (BChE) relates to antioxidant activity?

7.       The FRAP, DPPH and ABTS methods differ in their mechanism of action. Please comment whether there is any relationship observed in the activity of the plant material in relation to a specific mechanism of action

8.       Table 5 cites reference 32, but there is no activity determination method; the same for ref. 55

Author Response

Manuscript ID: ddc-2604855

We have now revised the manuscript in accordance with the comments from the reviewer.

The point by point changes are as follows:

The logical work arrangement should be changed. First, the chemistry should be described (chapter 4) and then the activity (chapter 3). I would even suggest mentioning the traditional use at the end. The review paper is about scientific proof of the effect of the plant material.

Reply: The changes have been amended as suggested. The changes have been colored red with yellow stripes within the manuscript.

  1. Figure 1 and Table 2 should be combined. Table 2 does not systematize and divide the compounds, e.g. into phenolic acids, flavonoids, etc. in the appropriate order. There is a lot of chaos that makes it difficult to understand the work. I don't understand whether their listing is accidental or whether they were listed in order of the highest content in the plant material?

Reply: Figure 1 and Table 2 have now been combined as proposed. Moreover, the compounds have now been divided into classes, such as, terpenoids, flavonoids, etc.

  1. Chapter 4 lacks information about the content of individual compounds, or even suggestions about which compounds may contain the most.

Reply: Information about the content of individual compounds has now been incorporated wherever possible in the main manuscript.

  1. Many MIC values are given in section 5.2. They make reading very difficult and it's hard for me to understand what they refer to. I suggest changing the form of presenting this data.

Reply: We agree with the reviewer’s inquisitiveness. We have now improved the form of presenting the data in a more summarized way. The changes have been colored red with yellow stripes.

  1. In chapter 5.3, the authors refer to the Folin-Ciocalteu method, which assesses the total polyphenol content. This is not a method that provides antioxidant activity. There is a high correlation between TPC and activity, but there was a large factual error in the work.

Reply: We agree with the reviewer’s apprehension. The error has now been corrected as suggested.

  1. Are you sure the inhibition of acetylcholinesterase (AChE) and butyrylcholinesterase (BChE) relates to antioxidant activity?

Reply: We agree with the reviewer’s apprehension. The change has been amended.

  1. The FRAP, DPPH and ABTS methods differ in their mechanism of action. Please comment whether there is any relationship observed in the activity of the plant material in relation to a specific mechanism of action

Reply: The difference in mechanism of action in FARP, DPPH and ABTS methods has now been discussed and incorporated at the appropriate place of the main text. The changes have been colored red with yellow stripes.

  1. Table 5 cites reference 32, but there is no activity determination method; the same for ref. 55

Reply: The suggested corrections regarding the determination method have been implemented in the main text. The changes have been colored red with yellow stripes.

Round 2

Reviewer 2 Report

Accept in present form